# Oxidation State in Peritoneal Dialysis in Patients with Type 2 Diabetes Mellitus

**DOI:** 10.3390/ijms24032669

**Published:** 2023-01-31

**Authors:** Leonardo Pazarín-Villaseñor, Yessica García-Salas, Francisco Gerardo Yanowsky-Escatell, Fermín Paul Pacheco-Moisés, Jorge Andrade-Sierra, Tannia Isabel Campos-Bayardo, Daniel Román-Rojas, Andrés García-Sánchez, Alejandra Guillermina Miranda-Díaz

**Affiliations:** 1Nephrology Service, Civil Hospital of Guadalajara “Dr. Juan I Menchaca”, Guadalajara 44280, Jalisco, Mexico; 2Nephrology Specialty, Regional General Hospital No. 46 of the IMSS, Guadalajara 44910, Jalisco, Mexico; 3Department of Health Sciences-Illness as an Individual Process, University Center of Tonala, University of Guadalajara, Guadalajara 45425, Jalisco, Mexico; 4Department of Chemistry, University Centre for Exact and Engineering Sciences, University of Guadalajara, Guadalajara 44430, Jalisco, Mexico; 5Department of Physiology, University Center of Health Sciences, University of Guadalajara, Guadalajara 44360, Jalisco, Mexico

**Keywords:** diabetes mellitus, oxidants, antioxidants, oxidative stress, peritoneal dialysis, ESRD

## Abstract

End-stage renal disease (ESRD) progression is closely related to oxidative stress (OS). The study objective was to determine the oxidant and antioxidant status in peritoneal dialysis (PD) patients with type 2 diabetes mellitus (DM). An analytical cross-sectional study from the PD program was carried out with 62 patients, 22 with and 40 without DM. Lipoperoxides (LPO) levels in patients with DM, 3.74 ± 1.09 mM/L, and without DM, 3.87 ± 0.84 mM/L were found to increase compared to healthy controls (HC) 3.05 ± 0.58 mM/L (*p* = 0.006). The levels of the oxidative DNA damage marker (8-OH-dG) were found to be significantly increased in patients with DM, 1.71 ng/mL (0.19–71.92) and without DM, 1.05 ng/mL (0.16–68.80) front to 0.15 ng/mL (0.15–0.1624) of HC (*p* = 0.001). The antioxidant enzyme superoxide dismutase (SOD) activity was found to be significantly increased in patients with DM, 0.37 ± 0.15 U/mL, and without DM, 0.37 ± 0.17 compared to HC, 0.23 ± 0.05 U/mL (*p* = 0.038). The activity of the enzyme glutathione peroxidase (GPx) showed a significant increase (*p* < 0.001) in patients with DM, 3.56 ± 2.18 nmol/min/mL, and without DM, 3.28 ± 1.46 nmol/min/mL, contrary to the activity obtained in HC, 1.55 ± 0.34 nmol/min/mL. In conclusion, we found an imbalance of oxidative status in patients undergoing PD with and without DM through the significant increase in LPO oxidants and the marker of oxidative damage in DNA. The activity of the antioxidant enzymes SOD and GPx were significantly increased in patients with and without DM undergoing PD, possibly in an attempt to compensate for the deregulation of oxidants. Antioxidant enzymes could be promising therapeutic strategies as a complement to the management of chronic kidney diseases.

## 1. Introduction

Chronic kidney disease (CKD) is one of Mexico’s principal public health challenges. The number of patients with CKD has increased about to the increased prevalence of metabolic diseases such as obesity, high blood pressure, and type 2 diabetes mellitus (DM) [1]. CKD is defined by estimated glomerular filtration rate (eGFR) < 60 mL/min/1.73 m^2^ or one or more markers of persistent kidney damage for three months; albuminuria (albumin excretion rate ≥ 30 mg/day, albumin-creatinine ratio (ACR)  ≥ 30 mg/g), urinary sediment abnormalities, electrolyte abnormalities, tubular disorders, abnormalities detected by histology, and structural abnormalities detected by imaging or history of renal transplant (RT) [2]. The prevalence of CKD varies between populations [3]. In Mexico, it is estimated that there are 12.8 million patients with type 2 DM [4]. Hyperglycemia is the primary determinant of diabetic nephropathy (DN) [5]. ND represents ~30–40% of end-stage renal diseases (ESRD) that require renal replacement therapy (RRT) and that, together with cardiovascular disease (CVD), are the leading causes of morbidity and mortality in the population [6].

The primary pathogenic mechanisms for the appearance of macro and microvascular complications of DN may be similar to those produced by reactive oxygen species (ROS) as common denominators of various signaling pathways [7,8,9]. ROS include molecular O_2_ and derivatives; superoxide anion (O_2_^−^), hydroxyl radical (HO**^.^**), hydrogen peroxide (H_2_O_2_), peroxynitrite (ONOO−), hypochlorous acid (HOCl), nitric oxide (NO), and lipid radicals. Many ROS possess unpaired electrons (free radicals). Excessive amounts of ROS oxidize several tissue biomolecules, such as DNA, proteins, carbohydrates, and lipids, with the capacity to produce oxidative stress (OS) [10]. OS is a common factor that couples hyperglycemia with vascular complications through two mechanisms: metabolic modifications of target tissue molecules and alterations in renal hemodynamics [11]. OS is prevalent in CKD and persists even after kidney transplantation [12]. Although ROS are as important as physiological defense mechanisms in the body, it results in tissue damage when over a generation of oxidants. OS is of paramount importance in determining the state of chronic inflammation in patients undergoing peritoneal dialysis (PD). On the other hand, choosing antioxidants is an emerging strategy to counteract OS in patients with DN on PD with the potential to preserve peritoneal function [13,14]. PD patients manifest excessive oxidative stress (OS) compared to the general population mainly due to the composition of the PD solution (high glucose content, low pH, high osmolality, increased lactate concentration, and products of glucose breakdown). However, PD could be considered a more biocompatible form of dialysis compared to hemodialysis (HD) [15]. Data on the behavior of OS in PD patients is still limited, although it is constantly growing.

The objective of the study was to determine the oxidative and antioxidant status in PD patients with type 2 DM.

## 2. Results

Forty patients without DM and twenty-two with Type 2 DM were included. Table 1 shows the demographic and biochemical data between patients with and without type 2 DM in the PD program. Patients with type 2 DM on PD were older in years (*p* < 0.001), more overweight (*p* < 0.039), had higher body mass index (BMI) (*p* < 0.015), lower serum albumin levels (*p* < 0.016), and higher glucose levels (*p* < 0.003). Patients without type 2 DM presented a greater increase in urea (*p* < 0.035), urea nitrogen (*p* < 0.030), creatinine (*p* < 0.001), phosphorus (*p* < 0.024), chloride (*p* < 0.038), potassium (*p* < 0.043) and had more arterial hypertension (*p* = 0.022). Albumin levels in all patients included in the study were decreased compared to normal levels. Triglycerides, cholesterol, and time on PD were similar between patients with and without type 2 DM included in the PD program.

Table 1 demographic and biochemical data of patients on PD with and without type 2 DM. Patients with type 2 DM on PD were older, more overweight, and had a higher body mass index (BMI), lower serum albumin levels, and higher glucose levels. Patients without type 2 DM presented a more significant increase in urea, urea nitrogen, creatinine, phosphorus, chloride, and potassium, and more arterial hypertension.

We performed a multivariate stepwise logistic regression analysis to determine the independence of the variables associated with DM. The multivariate logistic regression model showed the factors independently associated with DM in patients with PD are age, creatinine, albumin, and chloride.

### 2.1. Oxidants: Lipoperoxides, 8-Iso Prostaglandin F2α (8-Isoprostanes (8-IP), Nitric Oxide

LPO levels in patients with type 2 DM (3.74 ± 1.09 mM/L) and without type 2 DM (3.87 ± 0.84 mM/L) were found to be the significantly increased front the levels obtained in the healthy controls (HC), 3.05 ± 0.58 mM/L (*p* = 0.006). 8-IP levels were similar in patients with type 2 DM, 26.91 ± 7.42 pg/mL, without type 2 DM, 25.44 ± 6.76 pg/mL, and HC, 22.52 ± 11.77 pg/mL. NO metabolites (Nitrites/Nitrates) were similar between patients with type 2 DM 188.68 ± 118.50 mM, without type 2 DM, 233.76 ± 97.74 mM, and HC with 197.97 ± 108.16 mM (Table 2).

### 2.2. Antioxidants: Superoxide Dismutase, Glutathione Peroxidase, Catalase, and Total Antioxidant Capacity

The activity of the antioxidant enzyme SOD was significantly increased in patients with type 2 DM, 0.37 ± 0.15 U/mL, and without type 2 DM, 0.37 ± 0.17, front to HC, 0.23 ± 0.05 U/mL (*p* = 0.038) as seen in Table 2.

GPx enzyme activity was significantly increased (*p* < 0.001) in patients with type 2 DM, 3.56 ± 2.18 nmol/min/mL, and without type 2 DM, 3.28 ± 1.46 nmol/min/mL compared to the activity found in HC, 1.55 ± 0.34 nmol/min/mL.

Catalase enzyme activity was similar in patients without type 2 DM, 20.29 ± 1.01 mM, type 2 DM, 20.22 ± 0.67 mM, and HC, 20.03 ± 0.48 mM.

In the determination of TAC levels, similar results were obtained between patients with type 2 DM, 2.47 ± 0.53 mM, without type 2 DM, 2.64 ± 0.48 mM, and HC, 2.62 ± 0.52.

### 2.3. Markers of Oxidative Damage to DNA

A significant increase was found in the levels of the marker for oxidative damage to DNA (8-OH-dG) in PD patients with type 2 DM, 1.71 ng/mL (0.19–71.92), and without type 2 DM in the PD program, 1.05 ng/mL (0.16–68.80) front 0.15 ng/mL (0.15–0.1624) obtained in HC (*p* = 0.001). However, DNA repair enzyme levels were similar in patients with type 2 DM, without type 2 DM, and HC.

Table 2 oxidants and antioxidants in PD patients with and without type 2 DM front to healthy control. The significant increase in oxidants (LPO and the marker of oxidative damage to DNA) can be observed by the rise in the antioxidant activity of SOD and GPx without the DNA repair enzyme undergoing modifications.

Table 3 was adapted according to the main objective of this study; oxidant/antioxidant status in patients in PD with and without type 2 DM. This table shows that the peritoneal transport type correlates with increased DNA oxidative damage in non-diabetic patients. In DM, a positive correlation is observed with the concentration of DNA repair enzyme.

## 3. Discussion

Only patients in PD were included. The increase in OS occurs from the early stages of CKD, progresses with the deterioration of renal function, and is further aggravated by HD due to the bio incompatibility of the method. Compared to HD, PD is a more biocompatible dialysis modality, characterized by a significantly reduced, but still high OS status [16]. CKD patients in the PD program with type 2 DM had higher glucose levels, older age, higher BMI, and lower albumin levels. In this regard, serum albumin <3.8 g/dL is considered a biomarker used to diagnose protein-energy wasting [17,18]. In patients starting the RRT program. A decrease of 1 g/dL in serum albumin levels is associated with increased mortality risk [19]. Malnutrition has been reported in approximately one-third of ESRD patients undergoing PD. Factors influencing malnutrition in chronic dialysis patients are primarily due to poor nutritional intake, increased protein catabolism, and metabolic acidosis [20]. All patients had hypoalbuminemia compared to the average values; even hypoalbuminemia was lower in patients with type 2 DM. Studies have shown that ESRD patients with malnutrition have a higher hospitalization rate and are more susceptible to infections, fatigue, poor response to nutrition, and higher mortality [21]. Hypoalbuminemia in PD patients may result from the combined effects of elevated protein loss, malnutrition, inflammation, overhydration, and comorbidities [22]. The patients included in the study with type 2 DM have higher risks of secondary complications of diabetes and mortality. The importance of determining the albumin levels of patients included in dialysis programs should be highlighted. Serum albumin determination is a fundamental and straightforward biochemical method that determines the nutritional status of ESRD patients in PD [23].

Age, DM, and dialysis modality are factors that affect various functions of dialysis patients. It is recommended to pay special attention to patients with these characteristics to implement specific interventions with the aim of improving the functions of patients with PD. The older population always presents a higher frequency of DM, this is explained by the higher frequency of chronic degenerative diseases (DM, hypertension, dyslipidemia, etc.) as age advances, and as is known the greater frequency of mentioned comorbidities influences the worsening of the oxidative state in PD. Younger age and DM are significant risk factors. Younger patients often present psychosocial factors (lower socioeconomic level, less adherence to treatment, and less adherence to periodic assessments), almost constant characteristics of the disadvantaged population that is evaluated in this paper.

Dyslipidemia is generally characterized by high levels of triglycerides, low-density lipoprotein cholesterol (LDL-c), total cholesterol (TC), or low levels of high-density lipoprotein cholesterol (HDL-c). Dyslipidemia is a common atherogenic risk factor for CVD in CKD patients undergoing dialysis [21]. All patients included in the study with and without Type 2 DM had normal triglycerides and cholesterol levels. The presence of OS is of primary importance for the persistence of the state of chronic inflammation and fibrosis of the peritoneum in patients undergoing PD. The composition of the PD solution (low pH, high osmolality, higher concentration of lactate and glucose degradation products) seems to be actively involved in the accumulation of oxidative products [22]. Huh et al. suggest that the expansion of glucose degradation molecules in peritoneal solutions can trigger the appearance and development of OS [23]. OS occurs in all CKD patients and worsens as kidney function declines.

Lipid peroxidation appears to increase more during chronic hemodialysis. Although lipid peroxidation is also constant in PD [24]. The patients with and without type 2 DM increased LPO front the levels obtained in HC subjects.

8-Iso-PGF2α is derived from the oxidation of arachidonic acid produced by ROS stimulation [25]. Measurement of 8-iso-PGF2α is usually the approach to investigate OS. Urinary 8-iso-PGF2α excretion has become an OS assay used clinically in different vascular disease risk settings [26]. The contribution of the search for the urinary excretion of 8-iso-PGF2α has not been clarified so far [27]. The determination of 8-IP was performed in the serum of patients with and without type 2 DM undergoing PD. However, no alterations in this marker were found. In the present study, it would have been interesting to analyze the marker in the PD liquid.

ROS are generated through the processes of nicotinamide adenine dinucleotide phosphate (NADPH) oxidase, which reduces molecular O_2_ to O_2_^−^. The O_2_^−^ anion is converted to H_2_O_2_ by the enzyme SOD [28]. The O_2_^−^ anion reacts with NO to produce ONOO− and thus creates nitrosative stress [28]. H_2_O_2_ reacts with intracellular iron to form the H**^.^** [29]. H_2_O_2_ is catalyzed to hypochlorous acid (HOCl) in the presence of chloride ions by myeloperoxidase activity [30]. Uremic toxins can increase ROS generation and produce oxidation of macromolecules; nucleic acids, carbohydrates, lipids, and proteins [31]. OS induces endothelial dysfunction and atherosclerosis progression by reducing NO availability [32].

The NO levels in HC and patients with and without type 2 DM undergoing PD were similar, possibly because the time of inclusion in the PD program was short. In PD patients, NO bioavailability is associated with increased peritoneal membrane calcification, increased vascular endothelial growth factor (VEGF) activity, and increased accumulation of advanced glycation end products (AGEs) [33,34]. The increase in NO production participates as a pro-oxidant marker with the capacity to produce endothelial damage. The overproduction of NO may be the basis of the structural alterations in the peritoneal membrane in patients undergoing chronic PD [35]. OS due to ROS overproduction and impaired antioxidant defense mechanisms have been suggested as possible contributing factors to the pathogenesis of atherosclerosis in CKD patients included in this document [36].

Among the properties of ROS, the circulating level of 8-OHdG reflects the breakdown of DNA by oxidation. The 8-OHdG is a solid and sensitive marker of OS [37]. Circulating serum concentrations of 8-OHdG have increased in patients with cancer, chronic hepatitis, DM, heart disease, and ESRD on dialysis. OS assessed by the 8-OHdG marker could be an independent predictor of all-cause mortality in dialysis patients [38]. In the present study, we found a significant increase in oxidative DNA damage marker levels in PD patients with and without type 2 DM, making them vulnerable to premature death. OS may even be responsible for developing arterial stiffness in patients without and controlled type 2 DM. OS management may be necessary to prevent cardiovascular diseases in patients in the PD program [39]. The marker of oxidative DNA damage in type 2 DM appears to be triggered by increased glucose. Hyperglycemia promotes glucose oxidation and protein glycosylation and impairs DNA repair through DNA cleavage and ROS generation, leading to increased OS [40]. The repair of oxidative damage to DNA is essential to maintain the integrity of the genetic material, prevent mutagenesis, and decrease the damage caused by ROS. The hOGG1 enzyme is responsible for identifying and repairing oxidative DNA damage through base excision mechanisms [41]. High pro-inflammatory cytokines and ROS levels can decrease hOGG1 enzyme activity, especially in ESRD patients [41]. In the present study, the repair enzyme hOGG1 was similar between patients with and without type 2 DM, suggesting its inactivity or down-regulation due to the increased expression of the oxidative DNA damage marker. Little information is available in the literature on the behavior of the DNA repair enzyme in ESRD in dialysis with and without type 2 DM. The present study also provides scarce information in this regard, which makes it necessary to clarify in the future, with a more significant number of patients, the serum levels and the PD output fluid of the behavior of the DNA repair enzyme.

Antioxidant defense mechanisms are critical for protection against oxidative damage caused by free radicals. The endogenous antioxidant system consists of enzymatic antioxidants such as the enzyme SOD, CAT, and GPx [42].

The function of the SOD family metalloenzymes is to catalyze dismutation reactions against ROS. Coordinated metals in the active site of the enzyme facilitate its classification into copper (Cu^+2^), zinc (Zn^+2^), SOD (Cu/Zn-SOD), and manganese SOD (Mn-SOD), iron SOD (Fe^+2^-SOD), and nickel SOD (Ni-SOD) [31]. SOD appears to be the first line of defense against free radicals derived from molecular O_2_. Mitochondria are the prominent ROS-producing organelles and the main targets of the SOD enzyme. When the cell is exposed to OS, SOD is rapidly overexpressed. It is essential to consider that all isoforms of the enzyme are usually found in the kidney [43]. In the present study, the significant increase in SOD activity was notable in patients with and without type 2 DM compared to the results obtained in HC. The increased activity of the SOD enzyme could be due to the compensatory effect of the enzyme in the face of OS imbalance to inhibit mitochondrial oxidative damage and avoid altering mitochondrial membrane stability to the accumulation of ROS and free radicals. Mn-SOD production and activity are increased to inhibit mitochondria oxidative damage, prevent mitochondrial permeability transition, and preserve mitochondrial membrane stability [44].

Glutathione is the main non-enzymatic free radical scavenger. Glutathione is crucial in cellular resistance against cellular and systemic oxidative damage [45]. GPx is an essential family of antioxidant enzymes produced primarily in the kidneys. GPx is found in large amounts in the smooth muscle cells of the renal arteries and the proximal and distal tubules [46]. GPx1 is expressed mainly in the mitochondria of the renal cortex; its deficiency can reduce body weight and exacerbate the endogenous-dependent decline in general cell function. Renal GPx1 isoform regulation is crucial in renal protection against OS [47]. In the present study, the significant increase in the activity of the GPx enzyme in the plasma of PD patients with and without type DM compared to the results obtained in HC was notable. The increased GPx activity could also suggest a compensatory effect in response to the imbalance of the oxidative state characterized by the increase in LPO and the marker of oxidative damage to DNA. The increase in the activity of the GPx enzyme contrasts with the document reported in 2009, in which the authors found a marked decrease (60%) in the GPx enzyme in dialysis patients [48]. The articles available in the scientific literature have reported a significant reduction in the antioxidant enzymes SOD, catalase, and GPx in children on dialysis, regardless of the dialysis modality. On the contrary, other authors report increased enzymatic activity in erythrocytes of the enzymes SOD and GPx in patients with HD and PD [49,50].

The catalase enzyme is found primarily in peroxisomes and is abundant in the liver, lungs, and kidneys [51]. Catalase is distributed mainly in the cytoplasm of the proximal tubules of the renal juxtamedullary cortex and is less expressed in the proximal tubules of the superficial cortex. Catalase is not present in the glomeruli, distal tubules, loop of Henle, or collecting ducts [52]. Catalase levels were similar between HC and patients with and without type 2 DM included in the PD program, possibly due to the preferential localization of the enzyme within renal peroxisomes.

Although the determination of TAC levels is routinely performed in studies of OS in CVD, no data is available in the literature on the magnitude of the deterioration of antioxidant reserves in PD. There are even contradictory reports where normal levels of TAC are reported before submitting the patient to HD, decreasing levels after dialysis [53]. In the present study, serum TAC levels were determined immediately before the PD procedure, which is perhaps why TAC levels were similarly homogeneous between patients with and without type 2 DM compared to the results obtained from HC [54]. It was recently published that the TAC is increased in patients on dialysis (PD and HD), which could condition the increased risk of developing cardiovascular disease in patients with ERSD. No information was found in the literature on the behavior of the TAC scan in PD patients [55].

Antioxidants may support the preservation of peritoneal membrane function. The use of an angiotensin-converting enzyme inhibitor or angiotensin II receptor blocker helps preserve renal replacement therapy in patients with ESRD and maintains the integrity of the peritoneal membrane for a longer time in patients with PD. PD should be the initial dialysis modality in all patients with ESRD [56,57]. The results of the antioxidant activity obtained in the present study suggest an imbalance in relation to the oxidants. However, other studies would need to be carried out to determine the effect of some external antioxidants such as foods, vitamins, and statins with antioxidant effects, contemplating the adverse effects. In a recently published study, an increase in the activity of the SOD and GPx enzymes was found in patients undergoing renal transplantation with and without arterial hypertension managed with Enalapril, Losartan, or without antihypertensive treatment [58,59].

Given the findings obtained in the present study, we consider it worthwhile to determine the levels of all the oxidative state markers in the PD procedure’s exit liquid.

## 4. Materials and Methods

An analytical cross-sectional study was carried out on patients with CKD on PD. The patients have been hospitalized in the Nephrology Service of the “Juan I Menchaca” Civil Hospital in Guadalajara, Jalisco, Mexico. Patients with and without type 2 DM, adults >16 years, women, and men with a diagnosis of CKD with a minimum of one month and <6 months on continuous ambulatory PD who attended the peritoneal equilibrium test for the first time and agreed to sign the Informed Consent.

The type of peritoneal transport was obtained through the peritoneal equilibrium test and the patients were classified according to the result of the dialysate/plasma/creatinine ratio at 4 h into Low, Average Low, Average High, and PT High. Patients with an episode of peritonitis six months before the study, with catheter dysfunction, peritoneal fluid leak during the peritoneal equilibrium test procedure, and patients with systemic inflammatory disease (systemic lupus erythematosus, vasculitis, or other connective tissue) were not included as causes of CKD. The patients were determined; by age, weight, height, gender, blood pressure, hemoglobin, total cholesterol, HDL cholesterol, LDL cholesterol, triglycerides, creatinine, albumin and C-reactive protein, serum electrolytes, uric acid, ferritin, iron, and transferrin. To OS markers, 10 mL of blood was extracted, 5 mL with a 0.1% ethylenediaminetetraacetic (EDTA) tube, and another 5 mL in the dry tube. The blood was centrifuged at 10,000 rpm for 10 min at room temperature; supernatants were stored in aliquots at −80 °C until final processing. Then, 10 mL of extra blood from 10 blood donors (healthy control [HC]) was included and used to establish the standard value of the reagents. The characteristics of the population of healthy controls that participated in the study, were a blood donor population similar in age and free of any disease situation. 

### 4.1. Lipoperoxides (LPO)

Plasma LPO levels were measured using the FR22 assay kit (Oxford Biomedical Research Inc., Oxford, MI, USA^®^). The detection limit for this test was 0.1 nmol/mL. The chromogenic reagent reacts with malondialdehyde (MDA) and 4-hydroxy-alkenes to form a stable chromophore. Next, 140 µL of plasma was diluted with 455 µL of N-methyl-2-phenylindole acetonitrile (Reagent 1) with ferric iron in methanol. The samples were shaken, then 105 µL of 37% HCl was added, followed by incubation at 45 °C for 60 min and centrifugation at 12,791 rpm for 10 min. Then, 150 μL of the supernatant was added, and the absorbance at 586 nm was measured. The standard curve with known concentrations of 1,1,3,3-tetra methoxy propane in Tris-HCl was used. The intra-assay CV was 8.5%.

### 4.2. 8-Iso-Prostaglandin F2α (8-Isoprostanes (8-IP))

The Cayman Chemical Company^®^ (Ann Arbor, MI, USA) immunoassay reagent kit was used. The detection limit was 0.8 pg/mL. The assay was based on the principle of competitive binding between the 8-IP sample, the 8-IP acetylcholinesterase (AChE) conjugate, and the 8-IP tracer. Fifty µL of samples or standard was added to each well, and fifty µL of 8-IP AChE tracer was added to all wells except the blank and total activity wells. Fifty µL of 8-IP enzyme immunoassay antiserum was added to all wells except live and blank wells. Fifty µL of the 8-IP antiserum was added to all wells except total activity, non-specific binding, and blank wells. The plate was covered and incubated at 4 °C for eighteen h and then washed five times with buffer. The absorbance was read at 420 nm. The intra-assay CV was 12.5%.

### 4.3. Nitric Oxide (ON)

Before determining NO levels, serum samples were deproteinized by adding 6 mg of zinc sulfate to 400 μL of the example, vortexed for one min, and samples centrifuged at 10,000× *g* for ten min at 4 °C. The NO was determined by the colorimetric method and according to the kit (NO assay kit, User protocol 482650, Calbiochem^®^). Eighty-five µL of standard or sample was added to the plate wells, ten µL of nitrate reductase to each well, and ten µL of 2 mM NADH. The plate was shaken for twenty min at room temperature. Fifty μL of dye one was added and started briefly; then, fifty µL of dye two, and the samples were shaken for five min at room temperature. Finally, the plate was read at 540 nm in a spectrophotometer within the first twenty min of the procedure.

### 4.4. Antioxidants

#### 4.4.1. Superoxide Dismutase (SOD)

We followed the kit manufacturer’s instructions (SOD No. 706002, Cayman Chemical Company^®^, USA) for to detection of O_2−_ generated by the xanthine oxidase and hypoxanthine enzymes through the reaction of tetrazolium salts. The serum samples were diluted 1:2 in the sample buffer: two hundred µL of the radicals detector (1:400 dilution) and added ten µL of the example. After slow agitation, twenty µL of xanthine oxidase was added to the wells. Then, the microplate was incubated for twenty min at room temperature. The absorbency was read at 440 wavelengths of nm. The levels are reported in IU/mL.

#### 4.4.2. Catalase

Catalase was determined by kit according to the manufacturer’s instructions (Bioxytech Catalase-520^®^, cat. 21042, OXIS Int, Beverly Hills, CA, USA^®^), where thirty μL of diluted standard or sample and five hundred μL of the substrate (ten mM of H_2_O_2_) were added. Erythrocytes were incubated for one min at room temperature, and five hundred μL of stop reagent was added. The sample/standard was covered, mixed, and immersed, and twenty μL of each reagent was added to the tubes. Two milliliters of HRP/reactive chromogen were deposited, mixed in a water bath, and incubated for ten min. The absorbance was read at 520 nm.

#### 4.4.3. Glutathione Peroxidase (Plasma)

It was determined according to the commercial kit 703,102 from the manufacturer Cayman^®^. This method measures the decrease in absorbance caused by the reaction of an organic peroxide (tert-butyl hydroperoxide) with GPx of the sample studied in the presence of NADPH. A ninety-six well microplate is used; one hundred µL of assay buffer, fifty µL of the NADPH, glutathione, and glutathione reductase reagent mix, and twenty µL of the sample are added to each well. Twenty µL of cumene hydroperoxide is added to each well, and the absorbance obtained at 340 nm is recorded for the following 30, 60, 90, 120, 150, and 180 s. Subsequently, the rate of decrease in absorbance per min related to the concentration of NADPH consumed in the reaction is determined.

#### 4.4.4. Total Antioxidant Capacity (TAC)

TAC quantification was done following the (Total Antioxidant Power Kit, No. TA02.090130, Oxford Biomedical Research^®^). The serum samples and standards were diluted 1:40, and 200 μL were placed in each well of a microplate. Concentration was expressed as mM equivalents of Trolox (an analog of vitamin E). The duplicate standard intra-assay CV was 5.7%.

#### 4.4.5. Statistical Analysis

Normally distributed data were presented as mean ± standard deviation (SD). Based variables were displayed as median with interquartile range (IQR). Categorical variables were expressed as frequency and percentage. According to the type of data distribution, all demographic and PD-related characteristics were compared between diabetic and non-diabetic patients using Chi^2^, Student’s *t*-test, Mann–Whitney U-test, Kruskal–Wallis test with post hoc Dunn-Bonferroni test and Spearman’s correlation. Multivariate stepwise logistic regression analysis was used for adjusting confounding variables. Statistical analysis was performed using IBM SPSS v.18 software (Chicago, IL, USA). Any value of *p* ≤ 0.05 was considered statistically significant.

### 4.5. Ethical

The study was approved by the Local Ethics and Research Committee. Hospital Civil de Guadalajara “Dr. Juan I. Menchaca”. Guadalajara, Jalisco, México. (Jalisco Research Register no. 44/HCJIM-JAL/2017) on 2 February 2017.

## 5. Conclusions

PD provides better antioxidant protection than other types of RRT in CKD and should be considered the first treatment choice despite finding metabolic disorders. The imbalance between ROS formation and antioxidant system efficiency in the pathogenesis of cardiovascular complications that occur in ESRD with and without type 2 DM impacts patients undergoing PD. In the present study, we found an imbalance between the oxidants manifested by an increase in LPO, the marker of oxidative DNA damage, and the possible inactivity of the DNA repair enzyme. The increase in the antioxidants SOD and GPx activity was notable, suggesting a compensatory effect to protect mitochondrial integrity against the state of OS presented by patients with and without type 2 DM undergoing PD. Determining the concentrations of oxidant and antioxidant markers in serum, plasma, and PD outflow fluid from PD patients with type 2 DM could provide critical information for the best diagnostic and therapeutic approach in addition to traditional baseline management.

## Figures and Tables

**Table 1 ijms-24-02669-t001:** (a) Factors associated with type 2 DM in patients on PD in the multivariate model; (b) Demographic and biochemical data of patients on PD with type 2 DM.

(**a**)				
**Factors**	**OR**	**C. 95%**	** *p* **	
Age	1.243	1.003–1.541	0.047 *	
Glucose	1.087	1.005–1.177	0.038	
Albumin	0.001	<0.001–0.536	0.031 *	
(**b**)				
	**Healthy** **Control** **n-10**	**No DM** **n-40**	**DM** **n-22**	** *p* **
Gender				
Female n (%)	5	17 (42.5)	7 (31.82)	0.59
Male n (%)	5	23 (57.5)	15 (68.18)	
Smoking n (%)				
No smoking n (%)		26 (65)	17 (77.27)	0.61
Inactive smoking n (%)		12 (30)	4 (18.18)	
Active smoking n (%)		1 (2.5)	1 (4.55)	
Passive smoking n (%)		1 (2.5)	0 (0)	
Age years	32 (27–40)	27 (22–34)	53 (42–61)	<0.001 *
Weight kg	55 (55–70)	59.00 (49.70–67.75)	65.50 (57.75–75.50)	0.039 *
Height m	1.63 ± 0.1	1.62 ± 0.09	1.64 ± 0.12	0.42
BMI kg/m^2^	23.1 (20.0–24.1)	22.24 (19.59–24.37)	23.90 (22.20–27.97)	0.015 *
Hemoglobin g/dL		8.61 (7.88–11.10)	10.4 9 (8.40–11.32)	0.13
Iron µg/dL		60.81 ± 24.30	52.60 ± 18.79	0.17
Trasferrin mg/dL		182.08 ± 37.90	173.71 ± 35.60	0.4
TSAT%		25.66 ± 11.59	22.80 ± 11.19	0.84
Uric acid mg/dL		6.10 (5.13–7.00)	5.90 (5.38–6.1)	0.23
triglycerides mg/dL		132.20 ± 75.08	128.36 ± 59.42	0.84
Total Cholesterol mg/dL		160.50 (133.25–186.5)	168.00 (145.00–192.75)	0.52
HDL mg/dL		38.30 (34.63–46.68)	35.60 (29.78–47.88)	0.23
VLDL mg/dL		23.50 (14.25–33.00)	24.50 (13.75–36.25)	0.66
LDL mg/dL		100.85 (88.95–121.5)	109.40 (74.90–137.4)	0.34
Urea mg/dL		121.82 ± 43.85	98.65 ± 32.88	0.035 **
Ureic nitrogen mg/dL		57.17 ± 20.20	46.14 ± 15.37	0.030 **
Creatinine mg/dL		11.41 ± 3.72	8.21 ± 2.94	0.001 **
Glucose mg/dL		83.50 (79.00–93.00)	103.50 (85.00–131.25)	0.003
Albumin g/dL		2.87 ± 0.47	2.57 ± 0.45	0.016 **
Total proteins g/dL		5.79 ± 0.72	5.67 ± 0.622	0.50
Phosphorus mg/dL		4.35 (3.43–5.55)	3.65 (3.08–4.25)	0.024 *
Sodium mmol/L		140.00 (138.00–142.00)	138.50 (135.50–140.25)	0.13
Phosphorus calcium product mg/dL		40.60 (33.30–51.76)	36.39 (29.67–42.23)	0.09
Calcium mg/dL		8.70 (8.02–9.08)	8.50 (7.98–8.98)	0.49
Chlorine mmol/L		101.43 ± 4.50	98.86 ± 4.64	0.038 **
Potassium mmol/L		4.59 ± 0.64	4.14 ± 0.66	0.043 **
Magnesium mg/dL		2.25 ± 0.38	2.12 ± 0.32	0.18
Peritoneal Transport Type				
Low n (%)		11 (27.5)	3 (13.6)	
Average Low n (%)		11 (27.5)	5 (22.7)	<0.001 ^ŧ^
Average High n (%)		9 (22.5)	7 (31.8)	
High n (%)		9 (22.5)	7 (31.8)	
Dialysis dose		n (%)	n (%)	
2000 mL 1.5% every 6 h		30 (75)	11 (50)	0.26
2000 mL 2.25 to 2.50% % every 6 h		4 (10)	5 (22.7)	
2000 mL 2.25 to 2.50% every 6 h alternating		4 (10)	3 (13.6)	
2000 mL 1.5% every 6 h with a spare 2.5%		1 (2.5)	1 (4.5)	
2000 mL 1.5% every 8 h 2000 mL		1 (2.5)	2 (9.1)	

Values are mean ± SD, percent, or median (25th–75th percentile). BMI, body mass index; TSAT, transferrin saturation; HDL, high-density lipoprotein; LDL, low density lipoprotein; VLDL, very low density lipoprotein; ŧ Chi^2^ test, * Mann–Whitney U test; ** Student’s *t*-test. CI = confidence interval, BMI = Body mass index. Variables used in the regression: age, sex, body mass index, urea, blood urea nitrogen, creatinine, glucose, albumin, phosphorus, chlorine, potassium, peritoneal transport type.

**Table 2 ijms-24-02669-t002:** Oxidants and antioxidants in patients in PD with and without type 2 DM vs. healthy controls.

	Healthy Control	No-DM n-40	Type 2 DM n-22	*p*
Oxidants				
Lipoperoxides (micromol/L)	2.88 (2.7–3.20)	3.69 (3.09–4.44) ^a^	3.66 (3.10–4.0) ^a^	0.006 *
8-isoprostane (pg/mL)	22.52 ± 11.77	25.44 ± 6.76	26.91 ± 4.19	0.34
Nitric oxide (mM)	175.47 (114.17–302.68)	265.40 (148.04–314.77)	197.81 (61.76–283.42)	0.24
Antioxidants				
Superoxide dismutase (U/mL)	0.23 ± 0.48	0.37 ± 0.17 ^a^	0.37 ± 0.15 ^a^	0.038 *
Glutathione peroxidase (nmol/min/mL)	1.592 (1.24–1.81)	3.06 (2.29–3.66) ^a^	2.80 (2.13–4.39) ^a^	<0.001 *
Catalase (mM)	20. 00 (19.55–20.43)	20.17 (19.80–20.54)	20.22 (19.97–20.54)	0.54
Total Antioxidant Capacity (mM)	2.62 ± 0.52	2.64 ± 0.48	2.47 ± 0.53	0.43
Markers of oxidative damage to DNA				
8-OH-dG (ng/mL)	0.15 (0.15–0.1624)	1.05 (0.16–68.80) ^a^	1.71 (0.19–71.92) ^a^	0.001 *
hOGG1 (ng/mL)	0.03 (0.028–0.04)	0.04 (0.03–0.04)	0.04 (0.03–0.04)	0.68

8-OHdG = 8-Hydroxy-2’-deoxyguanosine, hOGG1 = 8-Oxoguanine glycosylase, ^a^, vs. Healthy control = HC. * Kruskal–Wallis test with post hoc Dunn-Bonferroni test.

**Table 3 ijms-24-02669-t003:** Spearman’s correlation coefficient between markers of OS and the type of PT with and without type 2 DM.

	No-DM		Type 2 DM	
	rho	*p*	rho	*p*
Lipoperoxides	−0.074	0.652	0.046	0.843
8-isoprostane	−0.041	0.800	−0.205	0.360
Nitric oxide	−0.183	0.258	0.332	0.131
Superoxide dismutase	−0.206	0.209	0.370	0.090
Glutathione peroxidase	−0.047	0.775	−0.405	0.061
Catalase	0.011	0.946	0.118	0.600
Total Antioxidant Capacity	0.161	0.320	−0.026	0.909
8-OH-dG	0.344 *	0.030	0.048	0.833
hOGG1	0.000	0.999	0.453 *	0.034

8-OHdG = 8-Hydroxy-2’-deoxyguanosine, hOGG1 = 8-Oxoguanine glycosylase. * Significant for Spearman correlation test *p* < 0.05.

## Data Availability

The database that supports the conclusions of this research work will be made available by the authors, upon express request and with the authorization of the Ethics and Research Committee.

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
