# Peer review of "Oxidation State in Peritoneal Dialysis in Patients with Type 2 Diabetes Mellitus"

_ijms, 2023, doi:10.3390/ijms24032669_

Round 1
Reviewer 1 Report
COMMENTS:
To the Editor
It was with interest that I read Pazarin-Villaseñor et al. manuscript.
The authors evaluated the oxidant and antioxidant status, evaluating markers not usually used in clinical practise, in 62 patients peritoneal dialysis (PD) patients with (n=22) and without (n=40) diabetes mellitus (DM).
PD patients are at high risk of serious and life-threatening illnesses, most notably cardiovascular diseases. There is an unmet medical requirement for biomarkers as tools to identify patients who are at the highest risk and to guide personalized interventions.
However, current approaches to patient monitoring are mostly limited to approximating delivered doses of dialysis and to measuring the membrane transport status.
11- In literature a correlation is described between type of peritoneal transport (PT) characteristics (low, low average, high average, and high PT) and significant changes in markers of inflammation, oxidative stress, and oxidative damage to DNA: however the authors do not report these data, that might be of interest in reinforcing their conclusions.
22- An imbalance of oxidative status was found in peritoneal dialysis patients suggesting anti-oxidant enzymes as a promising therapeutic strategies for the management of chronic kidney diseases but without having found differences from patients with or without DM: changes in the job title could be more correct by deleting with and without DM?
33- Another obeservation concerns the PD characteristics between the two grups of patients, with and without DM: a table with baseline characteristics might be useful as well as the number of hypertonic bags containing high glucose, potentially able to worsening the oxidative status.
44- There were significat differences in residual renal function, important factor influencing atherosclerosis/inflammation in PD patients?
55- Table 1, demographic data: patients without DM were significantly younger, probably with a more favourable nutritional staus (higher albumin, urea nitrogen, phosphorus and potassium) as correctly debated in Discussion section, and, surprisingly, more anemic than DM. Some clarification from the authors?
66- In literature are available more recent references?
Author Response
Reviewer 1.
To the Editor
It was with interest that I read Pazarin-Villaseñor et al. manuscript.
The authors evaluated the oxidant and antioxidant status, evaluating markers not usually used in clinical practise, in 62 patients peritoneal dialysis (PD) patients with (n=22) and without (n=40) diabetes mellitus (DM).
Comment. PD patients are at high risk of serious and life-threatening illnesses, most notably cardiovascular diseases. There is an unmet medical requirement for biomarkers as tools to identify patients who are at the highest risk and to guide personalized interventions.
However, current approaches to patient monitoring are mostly limited to approximating delivered doses of dialysis and to measuring the membrane transport status.
Answer. Done
Comment. 11- In literature a correlation is described between type of peritoneal transport (PT) characteristics (low, low average, high average, and high PT) and significant changes in markers of inflammation, oxidative stress, and oxidative damage to DNA: however the authors do not report these data, that might be of interest in reinforcing their conclusions.
Answer. All measurements were performed prior to performing a peritoneal equilibrium test, however, as peritoneal transport was not a primary objective of the study, this measurement was not included in the analysis. However, these data relevant to the association of OS and peritoneal transport have already been reported by our group in a previous study by Gutierrez-Prieto et al. “The Behavior of the Type of Peritoneal Transport in the Inflammatory and Oxidative Status in Adults Under Peritoneal Dialysis. Front Med (Lausanne). 2019;6:210. Published 2019 Sep 27. doi:10.3389/fmed.2019.00210”
Agregar en la discussion
Comment. 22- An imbalance of oxidative status was found in peritoneal dialysis patients suggesting anti-oxidant enzymes as a promising therapeutic strategies for the management of chronic kidney diseases but without having found differences from patients with or without DM: changes in the job title could be more correct by deleting with and without DM?
Answer. done
Comment. 33- Another observation concerns the PD characteristics between the two grups of patients, with and without DM: a table with baseline characteristics might be useful as well as the number of hypertonic bags containing high glucose, potentially able to worsening the oxidative status.
Answer. It can be added in table No.1 what refers to the concentration of dialysis bags of the population with diabetes compared to non-diabetic patients where no significance is observed (p = 0.26), this data can be added in table No. 1
|
PD fluid concentration |
DM |
No-DM |
total |
p= 0.26 |
|
1.5% |
13 |
31 |
44 |
|
|
2.5% |
5 |
4 |
9 |
|
|
1.5-2.5% |
5 |
4 |
9 |
|
Comment. 44- There were significat differences in residual renal function, important factor influencing atherosclerosis/inflammation in PD patients?
Answer. Residual renal function was not a variable evaluated in the present study.
Comment. 55- Table 1, demographic data: patients without DM were significantly younger, probably with a more favourable nutritional staus (higher albumin, urea nitrogen, phosphorus and potassium) as correctly debated in Discussion section, and, surprisingly, more anemic than DM. Some clarification from the authors?
Answer. Younger patients often present psychosocial factors (lower socioeconomic level, less adherence to treatment, and less adherence to periodic assessments), almost constant characteristics of the disadvantaged population that is evaluated at our hospital.
Comment. 66- In literature are available more recent references?
Answer. In the present study, the most relevant and most up-to-date bibliographic references on the specific subject in question were included.
Reviewer 2
Comments and Suggestions for Authors
In this cross-sectional study, the authors investigated oxidative stress in the peritoneal dialysis (PD) patients with and without type 2 diabetes mellitus (DM). Oxidant and antioxidant markers were measured and compared between groups: including PD patients with and without DM, and healthy control.
Comment. 1. In Table 1, there is no information about PD therapy at all. You should indicate them, such as PD vintage, modality (CAPD/APD), residual renal function, renal and peritoneal Kt/V, D/Pcre. You should also describe the information about type of PD solution, for example, whether neutral pH biocompatible or acidic PD solution, glucose concentration, and use of icodextrin solution.
Answer. Only patients with CAPD-based PD treatment participated in this study, so automated treatments were not included in the population evaluated. In addition, the PD population in our hospital only uses glucose-based PD solutions, the use of icodextrin solutions is not included as a PD treatment in our patients.
We added in Table No.1 what refers to the concentration of dialysis bags of the population with diabetes compared to non-diabetic patients where no statistical significance was observed (p=0.26). It should be noted that,variables concerning a residual renal function, renal and peritoneal Kt/V, D/Pcre, not were evaluated
Comment. 2. Characteristics of healthy control were not shown anywhere. I think several parameters including age, gender, and obesity will affect the results of oxidative stress status. You should show basic data of healthy control in Table 1.
Answer. If necessary, a brief text can be added in the material and methods section explaining the characteristics of the population of healthy controls that participated in the study, which were a blood donor population similar in age and free of any disease situation.
Comment. 3. In Table 2, it is hard to understand which two groups were significantly different. Please make it clear. Which markers were significantly different between PD patients with and without DM?
Answer. As mentioned in the lower part of Table 2, the comparison was made only between diabetic and non-diabetic patients vs. control group. If necessary, a comparison of diabetics and non-diabetics can be performed using Student's t test, Mann-Whitney U test, depending on the type and distribution of the required data.
Comment. 4. The results of multivariate analysis were not shown anywhere although significant differences were found between groups in Table 2. According to the description in the Statistical analysis section, multivariate analysis should have been performed.
Answer. As mentioned in statistical analysis section, the comparison was made only using according to the type of data distribution, all demographic and PD-related characteristics were compared between diabetic and non-diabetic patients using Chi2, Student's t test. Mann-Whitney U test and Spearman correlation. A multivariate analysis was not performed in this population, the statistical analysis section where multivariate analysis is mentioned should be omitted from the manuscript
Comment. 5. In Table 1, age is significantly different between DM and non-DM groups. Because aging is known as one of the factors affecting oxidative stress status, you should discuss its influence to results of your study in the Discussion section.
Answer. In the first paragraph of the discussion, we can add how the older population will always present a higher frequency of DM, this is explained by the higher frequency of chronic degenerative diseases (DM, Hypertension, Dyslipidemia, etc.) as age advances, and as is known the greater frequency of mentioned comorbidities influences the worsening of the oxidative state.
Comment. 6. In Table 2, total antioxidant capacity seemed to be equal between healthy control and PD patients although you discussed the compensatory elevation of SOD and GPx in PD patients. How do you explain it? In the discussion section, it seems to explain that the reason for this is that TAC values were measured before PD, but PD patients are performing CAPD daily and are in a steady state, I think.
Answer.
Comment. 7. In Table 3, correlations between oxidant/antioxidant stress markers and laboratory data in PD patients with/without type 2 DM were shown. I cannot understand the necessary of this table because the aim of this study is to determine the oxidant/ antioxidant status in PD patients with and without DM, respectively. In addition, several laboratory parameters shown in Table 1 were ignored in Table 3. Although, in the several pairs of oxidative stress markers and parameters, there are significant correlations between them, I think that confounding factors such as age might affect it. To confirm the laboratory factors significantly associated with oxidative stress markers, multivariate analysis will be needed.
Answer.
Comment. 8. In the Conclusion section, although you described that “PD provides better antioxidant protection than other types of RRT”, your data did not show it in this study. Also, whether the imbalance between ROS formation and antioxidant system efficiency “in the pathogenesis of cardiovascular complications” that occur in ESRD with and without type 2 DM impacts patients undergoing PD had not been clearly demonstrated in this study.
Answer.
Comment. 9. Smoking may affect oxidative stress status of the patients. Please add it in Table 1.
Answer. It can be added in table No.1 an analysis of the influence of smoking between both populations with and without DM can be added to table 1 of baseline characteristics, no significance is observed (p = 0.39), this data can be added in table No. 1
|
smoking |
DM |
No-DM |
total |
p= 0.39 |
|
no |
17 |
26 |
43 |
|
|
yes |
5 |
14 |
19 |
|
Comment. 10. In the introduction, the part of discussion about CKD was too long. Because the main participant of this study were PD patients, you should discuss more about PD or ESKD.
Answer. This point can be discussed in the discussion section
Comment. 11. In the Materials and Methods section, explanation of Glutathione peroxidase (plasma) was duplicated and that of total antioxidant capacity was lacking.
Answer. I apologize, it was a mistake to duplicate the GPx enzyme activity quantification method and omit the total antioxidant capacity methodology. Duplicate text was removed, and the description of the total antioxidant capacity method was added to the document
TAC quantification was done following the (Total Antioxidant Power Kit, No. TA02.090130, Oxford Biomedical Research®). The serum samples and standards were diluted 1:40, and 200 μL were placed in each well of a microplate. Concentration was expressed as mM equivalents of Trolox (an analog of vitamin E). The duplicate standard intra-assay CV was 5.7%.
Comment. 12. What is “peritoneal balance test” in the Materials and Methods section? Does it mean peritoneal equilibration test?
Answer. the correct term is "peritoneal balance test" instead of "peritoneal balance test", the above was a grammatical and/or translation error

Reviewer 2 Report
In this cross-sectional study, the authors investigated oxidative stress in the peritoneal dialysis (PD) patients with and without type 2 diabetes mellitus (DM). Oxidant and antioxidant markers were measured and compared between groups: including PD patients with and without DM, and healthy control.
1. In Table 1, there is no information about PD therapy at all. You should indicate them, such as PD vintage, modality (CAPD/APD), residual renal function, renal and peritoneal Kt/V, D/Pcre. You should also describe the information about type of PD solution, for example, whether neutral pH biocompatible or acidic PD solution, glucose concentration, and use of icodextrin solution.
2. Characteristics of healthy control were not shown anywhere. I think several parameters including age, gender, and obesity will affect the results of oxidative stress status. You should show basic data of healthy control in Table 1.
3. In Table 2, it is hard to understand which two groups were significantly different. Please make it clear. Which markers were significantly different between PD patients with and without DM?
4. The results of multivariate analysis were not shown anywhere although significant differences were found between groups in Table 2. According to the description in the Statistical analysis section, multivariate analysis should have been performed.
5. In Table 1, age is significantly different between DM and non-DM groups. Because aging is known as one of the factors affecting oxidative stress status, you should discuss its influence to results of your study in the Discussion section.
6. In Table 2, total antioxidant capacity seemed to be equal between healthy control and PD patients although you discussed the compensatory elevation of SOD and GPx in PD patients. How do you explain it? In the discussion section, it seems to explain that the reason for this is that TAC values were measured before PD, but PD patients are performing CAPD daily and are in a steady state, I think.
7. In Table 3, correlations between oxidant/antioxidant stress markers and laboratory data in PD patients with/without type 2 DM were shown. I cannot understand the necessary of this table because the aim of this study is to determine the oxidant/ antioxidant status in PD patients with and without DM, respectively. In addition, several laboratory parameters shown in Table 1 were ignored in Table 3. Although, in the several pairs of oxidative stress markers and parameters, there are significant correlations between them, I think that confounding factors such as age might affect it. To confirm the laboratory factors significantly associated with oxidative stress markers, multivariate analysis will be needed.
8. In the Conclusion section, although you described that “PD provides better antioxidant protection than other types of RRT”, your data did not show it in this study. Also, whether the imbalance between ROS formation and antioxidant system efficiency “in the pathogenesis of cardiovascular complications” that occur in ESRD with and without type 2 DM impacts patients undergoing PD had not been clearly demonstrated in this study.
9. Smoking may affect oxidative stress status of the patients. Please add it in Table 1.
10. In the introduction, the part of discussion about CKD was too long. Because the main participant of this study were PD patients, you should discuss more about PD or ESKD.
11. In the Materials and Methods section, explanation of Glutathione peroxidase (plasma) was duplicated and that of total antioxidant capacity was lacking.
12. What is “peritoneal balance test” in the Materials and Methods section? Does it mean peritoneal equilibration test?
Author Response
Reviewer 2
Comments and Suggestions for Authors
Comment. 1. In Table 1, there is no information about PD therapy at all. You should indicate them, such as PD vintage, modality (CAPD/APD), residual renal function, renal and peritoneal Kt/V, D/Pcre. You should also describe the information about type of PD solution, for example, whether neutral pH biocompatible or acidic PD solution, glucose concentration, and use of icodextrin solution.
Answer. Only patients with CAPD-based PD treatment participated in this study, so automated treatments were not included in the population evaluated. In addition, the PD population in our hospital only uses glucose-based PD solutions, the use of icodextrin solutions is not included as a PD treatment in our patients.
We added in Table No.1 what refers to the concentration of dialysis bags of the population with diabetes compared to non-diabetic patients where no statistical significance was observed (p=0.26). It should be noted that,variables concerning a residual renal function, renal and peritoneal Kt/V, D/Pcre, not were evaluated
Comment. 3. In Table 2, it is hard to understand which two groups were significantly different. Please make it clear. Which markers were significantly different between PD patients with and without DM?
Answer. Patients with and without DM had similar results. In the design of the document, we considered including a group of healthy subjects because we did not know whether patients with chronic kidney disease on PD without DM could have better results than patients with chronic kidney disease with DM. We found that there was no significant difference between patients with and without DM but there was a difference compared to healthy controls.
Table 2 was as follows
|
Table 2. Oxidants and antioxidants in patients in peritoneal dialysis with and without type 2 diabetes mellitus vs. healthy controls
|
||||
|
|
Healthy control |
No-DM n-40 |
Type 2 DM n-22 |
p |
|
Oxidants |
||||
|
Lipoperoxides (micromol/L) |
2.88 (2.7-3.20) |
3.69 (3.09-4.44)a |
3.66 (3.10-4.0)a |
0.006* |
|
8-isoprostane (pg/mL) |
22.52±11.77 |
25.44±6.76 |
26.91±.419 |
0.34 |
|
Nitric oxide (mM) |
175.47 (114.17-302.68) |
265.40 (148.04-314.77) |
197.81 (61.76-283.42) |
0.24 |
|
Antioxidants |
||||
|
Superoxide dismutase (U/mL) |
0.23±0.48 |
0.37±0.17 a |
0.37±0.15 a |
0.038** |
|
Glutathione peroxidase (nmol/min/mL) |
1.592 (1.24-1.81) |
3.06 (2.29-3.66) a |
2.80 (2.13-4.39) a |
<0.001* |
|
Catalase (mM) |
20. 00 (19.55-20.43) |
20.17 (19.80-20.54) |
20.22 (19.97-20.54) |
0.54 |
|
Total Antioxidant Capacity (mM) |
2.62±0.52 |
2.64±0.48 |
2.47±0.53 |
0.43 |
|
Markers of oxidative damage to DNA |
||||
|
8-OH-dG (ng/mL) |
0.15 (0.15-0.1624) |
1.05 (0.16-68.80) a |
1.71 (0.19-71.92) |
0.001* |
|
hOGG1 (ng/mL) |
0.03 (0.028-0.04) |
0.04 (0.03-0.04) |
0.04 (0.03-0.04) |
0.68 |
|
8-OHdG= 8-Hydroxy-2'-deoxyguanosine, hOGG1= 8-Oxoguanine glycosylase. a, vs. Health control= HC. *Kruskal Wallis test with post-hoc Dunn-Bonferroni test. |
||||
Comment. 4. The results of multivariate analysis were not shown anywhere although significant differences were found between groups in Table 2. According to the description in the Statistical analysis section, multivariate analysis should have been performed.
Answer. The results in Table 2 did not show significant differences between the study groups (diabetes and non-diabetes). However, Table 1 shows some values with significant differences between DM and non-DM groups. To determine the independence of the variables associated with DM. We performed a multivariate stepwise logistic regression analysis. The variables considered in the multivariate analysis are those that were significant in Table 1
|
Table 2a. Factors associated with type 2 DM in patients with PD in the multivariate model |
|||
|
Factors |
OR |
95% C.I. |
p |
|
Age |
1.105 |
1.019 - 1.20 |
0.02* |
|
Creatinine |
.600 |
0.36 – 0.996 |
0.049* |
|
Albumin |
.033 |
0.0025 - 0.44 |
0.01* |
|
Chlorine |
.612 |
0.43 - 0.88 |
0.01* |
|
CI=confidence interval, BMI=Body mass index. Variables used in the regression: age, sex, body mass index, urea, blood urea nitrogen, creatinine, glucose, albumin, phosphorus, chlorine, and potassium |
|||
The multivariate logistic regression model showed the factors independently associated with DM in patients with PD are age, creatinine, albumin, and chloride.
Comment. 5. In Table 1, age is significantly different between DM and non-DM groups. Because aging is known as one of the factors affecting oxidative stress status.
Answer. Age, DM, and dialysis modality are factors that affect various functions of dialysis patients. It is recommended to pay special attention to patients with these characteristics to implement specific interventions with the aim of improving the functions of patients with PD. Old age and DM are two important risk factors for increased inflammation in PD patients, although they are not contraindications for PD [ ]. Younger patients often present psychosocial factors (lower socioeconomic level, less adherence to treatment, and less adherence to periodic assessments), almost constant characteristics of the disadvantaged population that is evaluated in this paper
Silva MZC, Antonio KJ, Reis JMS, Alves LS, Caramori JCT, Vogt BP. Age, diabetes mellitus, and dialysis modality are associated with risk of poor muscle strength and physical function in hemodialysis and peritoneal dialysis patients. Kidney Res Clin Pract. 2021;40(2):294-303
Tsai CC, Lee JJ, Liu TP, Ko WC, Wu CJ, Pan CF, Cheng SP. Effects of age and diabetes mellitus on clinical outcomes in patients with peritoneal dialysis-related peritonitis. Surg Infect (Larchmt). 2013 Dec;14(6):540-6
Imtiaz R, Hawken S, McCormick BB, Leung S, Hiremath S, Zimmerman DL. Diabetes Mellitus and Younger Age Are Risk Factors for Hyperphosphatemia in Peritoneal Dialysis Patients. Nutrients. 2017;9(2):152
Comment. All patients had hypoalbum. You should discuss its influence to results of your study in the Discussion section.
Answer. Hypoalbuminemia in PD patients may result from the combined effects of elevated protein loss, malnutrition, inflammation, overhydration, and comorbidities. Hypoalbuminemia appears to be an important determinant of hydration status in PD patients. The malnutrition is common in PD patients and is associated with overhydration. Reference
Sikorska D, Olewicz-Gawlik A, Baum E, Pawlaczyk K, Oko A. The importance of hypoalbuminemia in peritoneal dialysis patients: Impact of gender. Adv Clin Exp Med. 2019;28(6):729-735
Comment. 6. In Table 2, total antioxidant capacity seemed to be equal between healthy control and PD patients although you discussed the compensatory elevation of SOD and GPx in PD patients. How do you explain it? In the discussion section, it seems to explain that the reason for this is that TAC values were measured before PD, but PD patients are performing CAPD daily and are in a steady state, I think.
Answer. It was recently published that the TAC is increased in patients on dialysis (PD and HD), which could condition the increased risk of developing cardiovascular disease in patients with ESRD. No information was found in the literature on the behavior of the TAC scan in PD patients [52]. Reference. Ahmadpoor P, Eftekhar E, Nourooz-Zadeh J, Servat H, Makhdoomi K, Ghafari A. Glutathione, glutathione-related enzymes, and total antioxidant capacity in patients on maintenance dialysis. Iran J Kidney Dis. 2009;3(1):22-7
Comment. 7. In Table 3, correlations between oxidant/antioxidant stress markers and laboratory data in PD patients with/without type 2 DM were shown. I cannot understand the necessary of this table because the aim of this study is to determine the oxidant/ antioxidant status in PD patients with and without DM, respectively. In addition, several laboratory parameters shown in Table 1 were ignored in Table 3. Although, in the several pairs of oxidative stress markers and parameters, there are significant correlations between them, I think that confounding factors such as age might affect it. To confirm the laboratory factors significantly associated with oxidative stress markers, multivariate analysis will be needed.
Answer. Table 3 was adapted according to the main objective of this study; oxidant/antioxidant status in patients in PD with and without type 2 DM. This table shows that the peritoneal transport type correlates with increased DNA oxidative damage in non-diabetic patients. In DM, a positive correlation is observed with the concentration of DNA repair enzyme
|
Table 3. Spearman's correlation coefficient between markers of OS and the type of PT with and without type 2 DM |
||||
|
|
No-DM |
Type 2 DM |
||
|
|
rho |
P |
rho |
p |
|
Lipoperoxides |
-0.074 |
0.652 |
0.046 |
0.843 |
|
8-isoprostane |
-0.041 |
0.800 |
-0.205 |
0.360 |
|
Nitric oxide |
-0.183 |
0.258 |
0.332 |
0.131 |
|
Superoxide dismutase |
-0.206 |
0.209 |
0.370 |
0.090 |
|
Glutathione peroxidase |
-0.047 |
0.775 |
-0.405 |
0.061 |
|
Catalase |
0.011 |
0.946 |
0.118 |
0.600 |
|
Total Antioxidant Capacity |
0.161 |
0.320 |
-0.026 |
0.909 |
|
8-OH-dG |
0.344* |
0.030 |
0.048 |
0.833 |
|
hOGG1 |
0.000 |
0.999 |
0.453* |
0.034 |
Comment. 8. In the Conclusion section, although you described that “PD provides better antioxidant protection than other types of RRT”, your data did not show it in this study. Also, whether the imbalance between ROS formation and antioxidant system efficiency “in the pathogenesis of cardiovascular complications” that occur in ESRD with and without type 2 DM impacts patients undergoing PD had not been clearly demonstrated in this study.

Round 2
Reviewer 2 Report
1. If you did not have data of residual renal function and Kt/V, you should describe them as limitation of this study, at least.
2. In table 1, you should show the results of peritoneal equilibration test. Although correlation between oxidative stress and peritoneal transport status might be investigated in Table 3, there were no data of peritoneal transport status anywhere in your manuscript.
3. The comment 2 from the reviewer 2 has not been answered. Characteristics of healthy control, at least age and gender should be described anywhere. The number of HC group should be shown in Table 2.
4. The aim of this study and conclusion were still unclear after revision. You described that “The study objective was to determine the oxidant and antioxidant status in peritoneal dialysis (PD) patients with and without diabetes mellitus (DM)” in the abstract, on the other hand, “The objective of the study was to determine the oxidative and antioxidant status in PD patients with type 2 DM” at the last of Introduction section. In addition, at the last of the Conclusion section, you argued that determining the concentrations of oxidant and antioxidant markers in PD patients with type 2 DM could provide critical information. Did you focus on oxidative stress of DM patients undergoing PD? Or, did you just describe the oxidative status in PD patients with and without DM? Carefully check for consistency of aim and conclusion of this study throughout the manuscript.
5. In the Method section, you described that “When significant differences in serum levels of OS markers were found between groups, multivariate analysis was performed to determine the interaction between DM and other factors associated with increased OS.” Don't you mean multivariate analysis with oxidative stress as the objective variable and clinical factors including DM (presence or absence) as explanatory variables? If multivariate analysis could not be performed, you should clearly describe the reason in the manuscript.
6. You seemed to perform multivariate stepwise logistic regression analysis to determine the independence of the variables associated with DM. I cannot understand the need of this analysis in this study. Also. Too many variables were included in the analysis despite the small number of patients in each group.
7. In Table 2, was not 8-OH-dG significantly increased in T2DM patients compared with HC?
Author Response
Reviewer 2-second revision
Comment. 1. If you did not have data on residual renal function and Kt/V, you should describe them as the limitation of this study, at least.
Answer. Since we do not have data on residual renal function and Kt/V, it was mentioned in the limitations of the study
Comment. 2. In table 1, you should show the results of peritoneal equilibration test. Although the correlation between oxidative stress and peritoneal transport status might be investigated in Table 3, there were no data of peritoneal transport status anywhere in your manuscript.
Answer.
We add the prevalence of the types of peritoneal transport in Table 1
|
Table 1. Demographic and biochemical data of patients on PD with type 2 DM |
|||
|
|
No DM n-40 |
DM n-22 |
p |
|
Gender |
|||
|
Female n(%) |
17(42.5) |
7(31.82) |
0.59 |
|
Male n(%) |
23(57.5) |
15(68.18) |
|
|
Smoking n(%) |
|
|
|
|
No smoking n(%) |
26 (65) |
17 (77.27) |
0.61 |
|
Inactive smoking n(%) |
12 (30) |
4 (18.18) |
|
|
Active smoking n(%) |
1 (2.5) |
1 (4.55) |
|
|
Passive smoking n(%) |
1 (2.5) |
0 (0) |
|
|
Peritoneal Transport Type |
|
|
|
|
Low n(%) |
11 (27.5) |
3 (13.6) |
<0.001ŧ |
|
Average Low n(%) |
11 (27.5) |
5 (22.7) |
|
|
Average High n(%) |
9 (22.5) |
7 (31.8) |
|
|
High n(%) |
9 (22.5) |
7 (31.8) |
|
|
Age years |
27 (22-34) |
53 (42-61) |
<0.001* |
|
Weight kg |
59.00 (49.70-67.75) |
65.50 (57.75-75.50) |
0.039* |
|
Height m |
1.62±0.09 |
1.64±0.12 |
0.42 |
|
BMI kg/m2 |
22.24 (19.59-24.37) |
23.90 (22.20-27.97) |
0.015* |
|
Hemoglobin g/dL |
8.61 (7.88-11.10) |
10.4 9 (8.40-11.32) |
0.13 |
|
Iron µg/dL |
60.81±24.30 |
52.60±18.79 |
0.17 |
|
Trasferrin mg/dL |
182.08±37.90 |
173.71±35.60 |
0.4 |
|
TSAT% |
25.66±11.59 |
22.80±11.19 |
0.84 |
|
Uric acid mg/dL |
6.10 (5.13-7.00) |
5.90 (5.38-6.1) |
0.23 |
|
triglycerides mg/dL |
132.20±75.08 |
128.36±59.42 |
0.84 |
|
Total Cholesterol mg/dL |
160.50 (133.25-186.5) |
168.00 (145.00-192.75) |
0.52 |
|
HDL mg/dL |
38.30 (34.63-46.68) |
35.60 (29.78-47.88) |
0.23 |
|
VLDL mg/dL |
23.50 (14.25-33.00) |
24.50 (13.75-36.25) |
0.66 |
|
LDL mg/dL |
100.85 (88.95-121.5) |
109.40 (74.90-137.4) |
0.34 |
|
Urea mg/dL |
121.82±43.85 |
98.65±32.88 |
0.035** |
|
Ureic nitrogen mg/dL |
57.17±20.20 |
46.14±15.37 |
0.030** |
|
Creatinine mg/dL |
11.41±3.72 |
8.21±2.94 |
0.001** |
|
Glucose mg/dL |
83.50 (79.00-93.00) |
103.50 (85.00-131.25) |
0.003 |
|
Albumin g/dL |
2.87±0.47 |
2.57±0.45 |
0.016** |
|
Total proteins g/dL |
5.79±0.72 |
5.67±0.622 |
0.50 |
|
Phosphorus mg/dL |
4.35 (3.43-5.55) |
3.65 (3.08-4.25) |
0.024* |
|
Sodium mmol/L |
140.00 (138.00-142.00) |
138.50 (135.50-140.25) |
0.13 |
|
Phosphorus calcium product mg/dL |
40.60 (33.30-51.76) |
36.39 (29.67-42.23) |
0.09 |
|
Calcium mg/dL |
8.70 (8.02-9.08) |
8.50 (7.98-8.98) |
0.49 |
|
Chlorine mmol/L |
101.43±4.50 |
98.86±4.64 |
0.038** |
|
Potassium mmol/L |
4.59±0.64 |
4.14±0.66 |
0.043** |
|
Magnesium mg/dL |
2.25±0.38 |
2.12±0.32 |
0.18 |
|
Dialysis dose |
n (%) |
n (%) |
|
|
2000 mL 1.5% every 6 h |
30 (75) |
11 (50) |
0.26 |
|
2000 mL 2.25 to 2.50% % every 6 h |
4 (10) |
5 (22.7) |
|
|
2000 mL 2.25 to 2.50% every 6 h alternating |
4 (10) |
3 (13.6) |
|
|
2000 mL 1.5% every 6 h with a spare 2.5% |
1 (2.5) |
1(4.5) |
|
|
2000 mL 1.5% every 8 h 2000 mL |
1 (2.5) |
2 (9.1) |
|
|
Values are mean ± SD, percent, or median (25th - 75th percentile). BMI, body mass index; TSAT, transferrin saturation; HDL, high-density lipoprotein; LDL, low density lipoprotein; VLDL, very low density lipoprotein; ŧ Chi2 test, * Mann-Whitney U test; **Student's t-test |
|||
Also, It is added the description of the peritoneal balance test in the methodology as follows:
The type of peritoneal transport was obtained through the peritoneal equilibrium test and the patients were classified according to the result of the dialysate/plasma/creatinine ratio at 4 h into Low, Average Low, Average High, and PT High
Comment. 3. The comment 2 from the reviewer 2 has not been answered. Characteristics of healthy control, at least age and gender should be described anywhere. The number of HC groups should be shown in Table 2.
Answer. These data are included in table 1
|
|
Table 1. Demographic and biochemical data of patients on PD with type 2 DM |
||||
|
|
Healthy Control n-10 |
No DM n-40 |
DM n-22 |
p |
|
|
Gender |
|
||||
|
Female n(%) |
5 |
17(42.5) |
7(31.82) |
0.59 |
|
|
Male n(%) |
5 |
23(57.5) |
15(68.18) |
||
|
Smoking n(%) |
0 |
|
|
|
|
|
No smoking n(%) |
|
26 (65) |
17 (77.27) |
0.61 |
|
|
Inactive smoking n(%) |
|
12 (30) |
4 (18.18) |
||
|
Active smoking n(%) |
|
1 (2.5) |
1 (4.55) |
||
|
Passive smoking n(%) |
|
1 (2.5) |
0 (0) |
||
|
Age years |
32 (27-40) |
27 (22-34) |
53 (42-61) |
<0.001* |
|
|
Weight kg |
55 (55-70) |
59.00 (49.70-67.75) |
65.50 (57.75-75.50) |
0.039* |
|
|
Height m |
1.63±0.1 |
1.62±0.09 |
1.64±0.12 |
0.42 |
|
|
BMI kg/m2 |
23.1 (20.0-24.1) |
22.24 (19.59-24.37) |
23.90 (22.20-27.97) |
0.015* |
|
|
Hemoglobin g/dL |
|
8.61 (7.88-11.10) |
10.4 9 (8.40-11.32) |
0.13 |
|
|
Iron µg/dL |
|
60.81±24.30 |
52.60±18.79 |
0.17 |
|
|
Trasferrin mg/dL |
|
182.08±37.90 |
173.71±35.60 |
0.4 |
|
|
TSAT% |
|
25.66±11.59 |
22.80±11.19 |
0.84 |
|
|
Uric acid mg/dL |
|
6.10 (5.13-7.00) |
5.90 (5.38-6.1) |
0.23 |
|
|
triglycerides mg/dL |
|
132.20±75.08 |
128.36±59.42 |
0.84 |
|
|
Total Cholesterol mg/dL |
|
160.50 (133.25-186.5) |
168.00 (145.00-192.75) |
0.52 |
|
|
HDL mg/dL |
|
38.30 (34.63-46.68) |
35.60 (29.78-47.88) |
0.23 |
|
|
VLDL mg/dL |
|
23.50 (14.25-33.00) |
24.50 (13.75-36.25) |
0.66 |
|
|
LDL mg/dL |
|
100.85 (88.95-121.5) |
109.40 (74.90-137.4) |
0.34 |
|
|
Urea mg/dL |
|
121.82±43.85 |
98.65±32.88 |
0.035** |
|
|
Ureic nitrogen mg/dL |
|
57.17±20.20 |
46.14±15.37 |
0.030** |
|
|
Creatinine mg/dL |
|
11.41±3.72 |
8.21±2.94 |
0.001** |
|
|
Glucose mg/dL |
|
83.50 (79.00-93.00) |
103.50 (85.00-131.25) |
0.003 |
|
|
Albumin g/dL |
|
2.87±0.47 |
2.57±0.45 |
0.016** |
|
|
Total proteins g/dL |
|
5.79±0.72 |
5.67±0.622 |
0.50 |
|
|
Phosphorus mg/dL |
|
4.35 (3.43-5.55) |
3.65 (3.08-4.25) |
0.024* |
|
|
Sodium mmol/L |
|
140.00 (138.00-142.00) |
138.50 (135.50-140.25) |
0.13 |
|
|
Phosphorus calcium product mg/dL |
|
40.60 (33.30-51.76) |
36.39 (29.67-42.23) |
0.09 |
|
|
Calcium mg/dL |
|
8.70 (8.02-9.08) |
8.50 (7.98-8.98) |
0.49 |
|
|
Chlorine mmol/L |
|
101.43±4.50 |
98.86±4.64 |
0.038** |
|
|
Potassium mmol/L |
|
4.59±0.64 |
4.14±0.66 |
0.043** |
|
|
Magnesium mg/dL |
|
2.25±0.38 |
2.12±0.32 |
0.18 |
|
|
Peritoneal Transport Type |
|
|
|
|
|
|
Low n(%) |
|
11 (27.5) |
3 (13.6) |
|
|
|
Average Low n(%) |
|
11 (27.5) |
5 (22.7) |
<0.001ŧ |
|
|
Average High n(%) |
|
9 (22.5) |
7 (31.8) |
|
|
|
High n(%) |
|
9 (22.5) |
7 (31.8) |
|
|
|
Dialysis dose |
|
n (%) |
n (%) |
|
|
|
2000 mL 1.5% every 6 h |
|
30 (75) |
11 (50) |
0.26 |
|
|
2000 mL 2.25 to 2.50% % every 6 h |
|
4 (10) |
5 (22.7) |
|
|
|
2000 mL 2.25 to 2.50% every 6 h alternating |
|
4 (10) |
3 (13.6) |
|
|
|
2000 mL 1.5% every 6 h with a spare 2.5% |
|
1 (2.5) |
1(4.5) |
|
|
|
2000 mL 1.5% every 8 h 2000 mL |
|
1 (2.5) |
2 (9.1) |
|
|
|
|
Values are mean ± SD, percent, or median (25th - 75th percentile). BMI, body mass index; TSAT, transferrin saturation; HDL, high-density lipoprotein; LDL, low density lipoprotein; VLDL, very low density lipoprotein; ŧ Chi2 test, * Mann-Whitney U test; **Student's t-test |
||||
Comment. 4. The aim of this study and conclusion were still unclear after revision. You described that “The study objective was to determine the oxidant and antioxidant status in peritoneal dialysis (PD) patients with and without diabetes mellitus (DM)” in the abstract, on the other hand, “The objective of the study was to determine the oxidative and antioxidant status in PD patients with type 2 DM” at the last of Introduction section. In addition, at the last of the Conclusion section, you argued that determining the concentrations of oxidant and antioxidant markers in PD patients with type 2 DM could provide critical information. Did you focus on oxidative stress of DM patients undergoing PD? Or, did you just describe the oxidative status in PD patients with and without DM? Carefully check for consistency of aim and conclusion of this study throughout the manuscript.
Answer. The objective of the study was to determine the oxidative and antioxidant status in PD patients with type 2 DM
Comment. 5. In the Method section, you described that “When significant differences in serum levels of OS markers were found between groups, multivariate analysis was performed to determine the interaction between DM and other factors associated with increased OS.” Don't you mean multivariate analysis with oxidative stress as the objective variable and clinical factors including DM (presence or absence) as explanatory variables? If multivariate analysis could not be performed, you should clearly describe the reason in the manuscript.
Answer. No significant differences were found in oxidative stress markers between the diabetic and non-diabetic groups. Therefore, a multivariate analysis was not performed.
The wording of the methodology is corrected as follows:
Normally distributed data were presented as mean ± standard deviation (SD). Bi-ased variables were displayed as median with interquartile range (IQR). Categorical variables were expressed as frequency and percentage. According to the type of data distribution, all demographic and PD-related characteristics were compared between diabetic and non-diabetic patients using Chi2, Student's t-test, Mann–Whitney U-test, Kruskal Wallis test with post-hoc Dunn-Bonferroni test and Spearman's correlation. Multivariate stepwise logistic regression analysis was used for adjusting confounding variables. Statistical analysis was performed using IBM SPSS v.18 software (Chicago, IL, USA). Any value of p≤0.05 was considered statistically significant
Comment. 6. You seemed to perform multivariate stepwise logistic regression analysis to determine the independence of the variables associated with DM. I cannot understand the need of this analysis in this study. Also. Too many variables were included in the analysis despite the small number of patients in each group.
Answer. We performed logistic regression to accurately determine the variables with significant differences from table 1 after adjusting for confounding variables. The confounders included in the analysis are those that were significant in Table 1. The significant parameters from the multivariate stepwise logistic regression analysis show the variables linked to diabetes mellitus in patients with peritoneal dialysis independently of the confounding factors.
Comment. 7. In Table 2, was not 8-OH-dG significantly increased in T2DM patients compared with HC?
Answer. We appreciate the observation and the previous ones. Indeed, 8-OH-dG was significantly increased in type 2 DM patients compared to HC. The annotation is added in table 2
Round 3
Reviewer 2 Report
Manuscript has been corrected.